# Exchange-of-Thought: Enhancing Large Language Model Capabilities through Cross-Model Communication

**Zhangyue Yin**$^\diamond$    **Qiushi Sun**$^\heartsuit$    **Cheng Chang**$^\diamond$
**Qipeng Guo**$^{\diamond\clubsuit}$    **Junqi Dai**$^\diamond$    **Xuanjing Huang**$^\diamond$    **Xipeng Qiu**$^{\diamond*}$

$^\diamond$School of Computer Science, Fudan University
$^\heartsuit$National University of Singapore $^\clubsuit$Shanghai AI Laboratory
{yinzy21,changc21,jqdai22}@m.fudan.edu.cn   qiushisun@u.nus.edu
{qpguo16, xjhuang, xpqiu}@fudan.edu.cn

## Abstract

Large Language Models (LLMs) have recently made significant strides in complex reasoning tasks through the Chain-of-Thought technique. Despite this progress, their reasoning is often constrained by their intrinsic understanding, lacking external insights. To address this, we propose Exchange-of-Thought (EoT), a novel framework that enables cross-model communication during problem-solving. Drawing inspiration from network topology, EoT integrates four unique communication paradigms: Memory, Report, Relay, and Debate. This paper delves into the communication dynamics and volume associated with each paradigm. To counterbalance the risks of incorrect reasoning chains, we implement a robust confidence evaluation mechanism within these communications. Our experiments across diverse complex reasoning tasks demonstrate that EoT significantly surpasses established baselines, underscoring the value of external insights in enhancing LLM performance. Furthermore, we show that EoT achieves these superior results in a cost-effective manner, marking a promising advancement for efficient and collaborative AI problem-solving.

> "Two heads are better than one."
>
> *–English Proverb*

## 1 Introduction

Large Language Models (LLMs) such as GPT-4 (OpenAI, 2023) are revolutionizing the field of Natural Language Processing (NLP) by utilizing vast training corpora and huge computational resources (Bai et al., 2022a; Ouyang et al., 2022; Chowdhery et al., 2022; Zhang et al., 2022; Touvron et al., 2023a, *inter alia*). Although LLMs achieve exemplary performance across a wide range of NLP tasks (Wei et al., 2022a; Chung et al., 2022), they consistently struggle to perform well in

---
$^*$ Corresponding author.

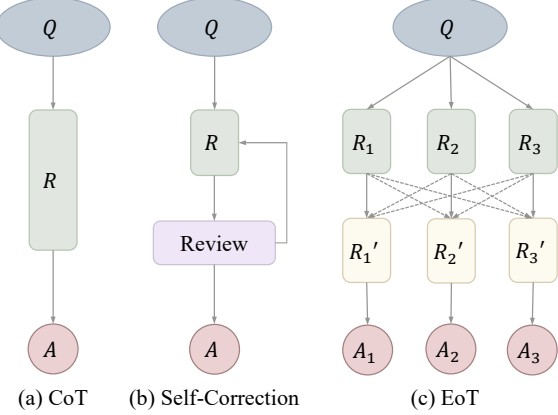

(a) CoT    (b) Self-Correction    (c) EoT

Figure 1: Comparison of CoT, Self-Correction, and EoT. Both CoT and Self-Correction rely on the model's innate abilities to generate and refine output, lacking external insights. EoT enhances the model's reasoning ability by incorporating the thoughts of other models as external insights.

reasoning tasks, and this limitation cannot be overcome solely by increasing the size of models (Rae et al., 2022; bench authors, 2023).

To overcome this shortcoming, Wei et al. (2022b) proposed chain-of-thought (CoT) prompting, which guides the model to generate a series of intermediate reasoning steps before reaching the final answer. At the same time, a series of self-correction methods (Madaan et al., 2023; Welleck et al., 2023; Shinn et al., 2023) have been proposed, which aim to iteratively improve the quality of answers by using the models' feedback on previous outputs (Bai et al., 2022b).

However, CoT and self-correction solely base on the model's own understanding and perspective of the question during the reasoning process. Recent studies (Huang et al., 2023; Valmeekam et al., 2023; Stechly et al., 2023) indicate that LLMs struggle to revise their responses without external feedback. This can be attributed to the model's complete dependence on internal representations to generate responses, which makes it difficult to overcome inherent limitations in capability (Yin et al., 2023).

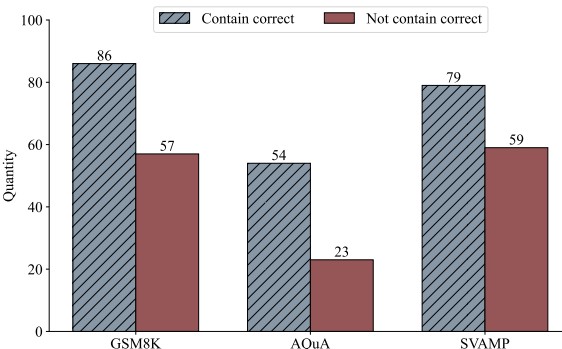

Figure 2: Pilot experiments on three reasoning datasets. The number of erroneous samples containing the correct answer is significantly higher than those not containing the correct answer.

Despite the undeniable importance of external insights (Yao et al., 2023), acquiring high-quality external insights remains a challenge. Wang et al. (2023c)'s research suggests that the single reasoning chain generated by CoT limits the model's reasoning performance. By increasing the temperature to sample diverse reasoning chains and selecting answers through majority voting, the model's reasoning performance can be further improved. However, when confronted with difficult questions, the model often yields a higher number of incorrect responses. In Figure 2, our analysis of correct and incorrect answers within erroneous samples from three reasoning datasets reveals that in most cases the model can deduce the correct answer.

In human society, the truth, even when held by a minority, can gain widespread acceptance and recognition through clear and persuasive communication (Le Bon, 1897). The correct reasoning of others can serve as high-quality external insights, which enrich and enhance our collective understanding. Thus, we propose Exchange-of-Thought (EoT), a novel framework that fosters cross-model communication during the problem-solving process. This initiative enables models to incorporate the reasoning of others as external insights.

Figure 1 contrasts EoT with CoT and self-correction methods, highlighting the unique approach of EoT in integrating external perspectives. Inspired by the principles of network topology (Bisht and Singh, 2015) and agent communication (Parsons and McBurney, 2003), we propose four communication paradigms: Memory, Report, Relay, and Debate. These paradigms are designed to facilitate the exchange of ideas and reasoning chains among models, enriching the problem-solving process with a diversity of insights. Fur-

thermore, we delve into the intricacies of each communication paradigm, analyzing the dynamics of information flow and the volume of communication. With the awareness that both correct and incorrect reasoning chains propagate within communications, we introduce confidence evaluation mechanisms that employ the analysis of answer variations to assess models' confidence levels. It is designed to mitigate the influence of erroneous reasoning, thereby ensuring the integrity and reliability of the problem-solving process.

Experiments across various complex reasoning tasks demonstrate that EoT significantly outperforms established strong baselines, underscoring the critical role of external insights in augmenting the capabilities of LLMs. We summarize our contributions as follows:

- We introduce Exchange-of-Thought (EoT), a pioneering framework for cross-model communication that incorporates external insights from other LLMs during problem-solving.

- We present and examine four communication paradigms coupled with a confidence evaluation mechanism that assesses model certainty through the variability of answers, mitigating the impact of incorrect reasoning.

- Experimental results on various complex reasoning tasks underscore the efficacy and cost-effectiveness of EoT, highlighting the significance of incorporating external insights and communication in problem-solving.

## 2 Related Work

### 2.1 Chain-of-Thought prompting in LLMs

Wei et al. (2022b) highlight that LLMs can manifest enhanced reasoning capabilities when being prompted by demonstrations with intermediate reasoning steps. This technique can effectively improve the performance of LLMs on complex reasoning tasks (Wei et al., 2022a; Kaplan et al., 2020). A series of strategies for enhancing CoT has been proposed to further improve the performance of LLMs. One such method is Program-aided language models (Gao et al., 2022; Chen et al., 2022), which aims to decouple reasoning and computation through program synthesis. Moreover, complex tasks can also be transformed into delegable sub-tasks through modular approaches (Khot et al., 2023). Choosing appropriate demonstrations can also enhance the performance of CoT (Li et al.,

2023a; Li and Qiu, 2023a). Notable among these, AutoCoT (Zhang et al., 2023b) uses an automated way to construct and sample diverse demonstrations. Active-Prompt (Diao et al., 2023) selects the most helpful samples for labeling based on the model's uncertainty in the outputs. Recently, Li and Qiu (2023b) employ a strategy of storing high-confidence thoughts as external Memory and retrieves these insights to aid the reasoning process.

## 2.2 Ensemble of Reasoning Paths

LLMs have the ability to explore multiple reasoning paths using techniques such as temperature adjustment and prompt sampling (Chu et al., 2023). Wang et al. (2023c) suggest that for complex questions, there may be several correct paths to approach a problem, leading to the proposal of Self-Consistency. This method replaces the greedy decoding strategy with the sampling of multiple reasoning paths and selecting the most consistent answer, resulting in significant performance improvements. Beyond that, Fu et al. (2023b) discover that prompts with higher reasoning complexity could achieve better performance in multi-step reasoning tasks, leading to the proposal of complexity-based prompting. While other methods, such as re-ranking (Thoppilan et al., 2022; Cobbe et al., 2021), have also been applied to select suitable reasoning paths, they often rely on heuristic or trained smaller models. Recently, Li et al. (2023b) sample different demonstrations and use step-by-step verification to filter out incorrect answers. However, obtaining step-level labels can be challenging, and using smaller models for judgment struggles to handle complex reasoning processes. In contrast, our method fully utilizes the communication and decision-making capabilities of LLMs to reach the final answer.

## 2.3 Reasoning Path Refinement

Although CoT (Wei et al., 2022b) effectively enhances the performance of LLMs in complex reasoning tasks, they remain susceptible to errors during the reasoning process, leading to incorrect answers (Bai et al., 2022b; Lyu et al., 2023). To mitigate this issue, starting from the model's own thoughts, Shinn et al. (2023) and Madaan et al. (2023) employ the LLMs' own outputs and past mistakes to refine the reasoning process. Yao et al. (2023) explore the synergies between reasoning traces and action plans. For numerical problems, Zheng et al. (2023) gradually guides models to the

correct answer by using previously generated answers as hints. With the aid of external knowledge, Wang et al. (2023a) introduce chain-of-knowledge prompting that employs evidence triples to curb the generation of unfactual and unfaithful answers. Taking model interactions into account, multi-agent debates (Du et al., 2023; Liang et al., 2023) have been introduced to enhance the factual accuracy of generated content and reduce fallacies and hallucinations. EoT differs from these efforts as we prioritize reasoning chains over simply providing answers. Our communication method involves multiple LLMs, which not only considers the historical information of a single LLM but also takes into account the interactive information among them.

## 3 Preliminary

Firstly, we define the current methods that use LLMs to solve problems. We denote a LLM with a parameter size of $\theta$ as $p_\theta$, and the sequence length as $t$, which includes tokens $[s_1, s_2, \ldots, s_t]$. The LLM predicts the next token based on the prior tokens in the sequence. The probability of the $s_i$ token is $p_\theta(s_i|s_1, s_2, \ldots, s_{i-1})$. Therefore, the probability of the whole sentence is $p_\theta(s) = \prod_{i=1}^{t} p_\theta(s_i|s_{\leq i-1})$.

**Standard prompting.** Standard prompting involves deriving an answer $a$ from a question $q$ using $p_\theta(a|q)$. To bolster the comprehension of LLM, an instruction prompt $I$ can be included before the question, resulting in $p_\theta(a|I, q)$. In-Context Learning, proposed by Brown et al. (2020), aims to improve LLMs performance by adding demonstrations $D = \{d_1, d_2, \ldots, d_n\}$ to the input, which can be expressed as $p_\theta(a|D, q)$.

**CoT prompting.** As identified by Wei et al. (2022b), the incorporation of intermediate reasoning steps can improve the proficiency of LLMs in tackling complex reasoning challenges. To facilitate this, a rationale $r_i$ is added to examples $d_i = \{q_i, r_i, a_i\}$ to guide the LLMs in explicitly generating reasoning steps. Kojima et al. (2022) propose that including "Let's think step by step" in the instruction $I$ can achieve the same effect, eliminating the need for hand-crafted examples.

**Self-Consistency** Self-Consistency proposed by Wang et al. (2023c), is a method that consolidates answers obtained from multiple independent reasoning chains. It returns the answer with the highest frequency of occurrence, represented as

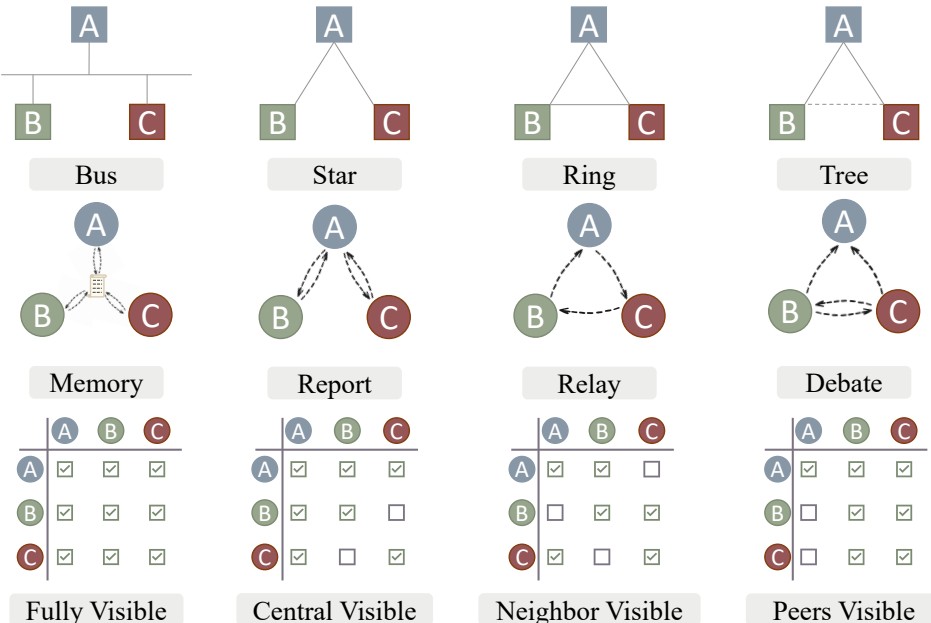

Figure 3: Correspondence between communication paradigms and network topologies. The top row depicts four network topologies. The second row correlates these with the corresponding communication paradigms. The bottom row offers an analysis of the communication volume associated with each mode. The horizontal axis represents the information that models can receive, while the vertical axis indicates the information that models can send.

$a = \mathrm{argmax}_{a_i} f(a_i)$, where $f(a_i)$ represents the frequency of occurrence of answer $a_i$. This approach enables the model to explore a wider array of reasoning paths, thereby enhancing its inferential capabilities. However, it remains constrained by the intrinsic limitations of LLMs' capabilities.

# 4 Methodology

To incorporate external insights, we introduce Exchange-of-Thought (EoT), a novel framework that facilitates communication and the exchange of ideas between models during the problem-solving process. This innovative approach leverages the communicative abilities of LLMs to promote the sharing of reasoning chains $r$ and answers $a$ among participating models, fostering a collaborative environment for thought and analysis. The implementation of EoT encounters three key challenges:

1. How to identify the appropriate counterparts for model communication?

2. What are the conditions for ceasing communication between models?

3. How to minimize the influence of incorrect reasoning during the communication process?

## 4.1 Communication Paradigm

Inspired by network topology (Bisht and Singh, 2015) and intelligent agent communication (Par-

sons and McBurney, 2003), we propose four communication paradigms to determine the counterparts for model communication. As illustrated in Figure 3, we propose Memory, Report, Relay, and Debate communication paradigms each corresponding to the Bus, Star, Ring, and Tree network topologies, respectively. Assume in $j$-th round of communication, given a set of LLMs $\{M\} = \{m_1, m_2, \ldots, m_n\}$, the model $m_i$ generates the corresponding rationale $r_i^{(j)}$ and the answer $a_i^{(j)}$ based on the $(r_K^{(j-1)}, a_K^{(j-1)})$, where $K$ is the set from which model $i$ can receive reasoning chains. In the first round, we use the CoT method proposed by Wei et al. (2022b) to generate $(r^{(1)}, a^{(1)}) \sim P_\theta(r^{(1)}, a^{(1)}|D, q)$.

**Memory** Under the Memory paradigm, all models record their rationale $r$ and answer $a$ in a logbook, which is fully visible from all models. This means that in the $j$-th round, any model, such as model $m_A$, can access the reasoning chains and answers from all models $(r_m^{(j-1)}, a_m^{(j-1)})$, $m \in \{M\}$. As depicted in Figure 3, this paradigm facilitates the fastest flow of information and also incurs the highest communication cost among all paradigms.

**Report** Under the Report paradigm, we designate model $m_A$ as the central node, which can obtain the rationale and answer from all other models $(r_m^{(j-1)}, a_m^{(j-1)})$, $m \in \{M\}\backslash\{m_A\}$. Both $m_B$ and

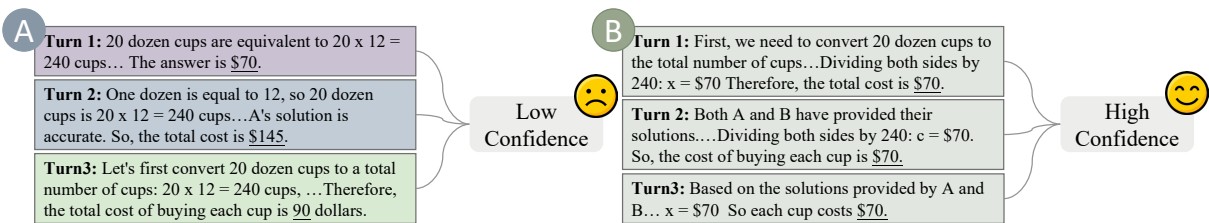

Figure 4: An illustrative comparison between a confident model and an unconfident model. Model A's answers fluctuate over three dialogue rounds, indicating uncertainty about the answer, while Model B consistently believes in a single answer.

$m_C$ only receive information from $m_A$ and do not interact with each other. Consequently, $m_A$ plays a pivotal role in the communication process. This paradigm also allows for rapid information flow, but it demands a higher capacity for processing and analysis from the central node.

**Relay**    Under the Relay paradigm, we order the models by number and connect them in a circle. Each node is capable of receiving information from the preceding node and transmitting its own information to the subsequent node. For example, in the $j$-th round, $m_A$ passes $(r_A^{(j-1)}, a_A^{(j-1)})$ to $m_C$ and receives $(r_B^{(j-1)}, a_B^{(j-1)})$ from the previous round of $m_B$. This distributed mode of communication can alleviate the demands on information processing capacity of a single node, yet it may result in slower information flow.

**Debate**    We have adapted the tree topology to devise the Debate paradigm. This paradigm permits leaf nodes to exchange information with each other, while parent nodes are solely responsible for aggregating information, meaning that information flow is directed upward from child to parent. As illustrated in Figure 3, $m_B$ and $m_C$, as child nodes, are able to communicate, whereas $m_A$, as a parent node, can only receive information from its children. This communication paradigm balances the model's information processing capacity with the speed of information flow.

## 4.2   Communication Volume

The last row of figure 3 displays the information that can be transmitted and received through different communication paradigms. The communication volume is measured by the number of messages received, assuming there are $n$ models participating in the communication, with each node transmitting its information from the previous round to the next.

In the Memory paradigm, every node receives information from all other nodes in the previous round, resulting in a communication volume of $n^2$. Any piece of information requires only one transmission to reach the corresponding node.

Under the Report paradigm, the central node processes $n$ pieces of information in each round, while the remaining $n-1$ nodes process 2 each. Therefore, the communication volume is $3n-2$, and any piece of information requires at most two transmissions via the central node to reach the corresponding node.

In the Relay paradigm, each node receives information from itself and the previous node from the last round, leading to a communication volume of $2n$. Node $i$ receives information from node $i-1$ in one transmission, while receiving information from node $i+1$ requires $n-1$ transmissions, giving an average propagation speed of $\frac{n-1}{2}$.

For the Debate paradigm, we assume the nodes form a binary tree with a height of $h = \lceil log_2(n+1) \rceil$. Each pair of child nodes has a communication volume of 2, and the parent node has a volume of 3, so a subtree composed of two children and one parent has a communication volume of 7. The number of such subtrees in a full binary tree is $2^{h-1} - 1$. Assuming $n$ is odd, the calculation for the communication volume $\mathcal{V}$ proceeds as follows:

$$\mathcal{V} = 7(2^{h-1} - 1 - \frac{2^h - 1 - n}{2}), \quad (1)$$

which simplifies to $\frac{7}{2}(n-1)$. The aggregation of information between two nodes, without considering other communications, requires at most $2(h-1)$ transmissions.

## 4.3   Termination Condition

Utilizing the model's current round output and the answers from previous rounds, we have devised two criteria for terminating communication: consistent output and majority consensus.

**Consistent Output Termination.** Inspired by Zheng et al. (2023), we implement a consistent output termination in EoT. The termination condition is triggered when the output of model $m_i$ in the $j$-th round is the same as the output in the $j-1$-th round, $a_i^{(j)} = a_i^{(j-1)}$. In this case, $m_i$ will stop receiving or sending information and exit the current communication.

**Majority Consensus Termination.** Du et al. (2023) observed that LLMs can converge on a consensus after several rounds of debate, suggesting that LLMs fine-tuned with reinforcement learning from human feedback (RLHF) (Ouyang et al., 2022) are more likely to reach an agreement. Inspired by this finding, we propose the termination condition of majority rule, where LLMs cease communication with each other once a majority of them reach an agreement. This approach serves as a global termination condition, distinguishing it from the consistent output termination, which acts as a cessation criterion on an individual model basis.

### 4.4 Confidence Evaluation

An intriguing aspect of human behavior is that individuals tend to be less susceptible to incorrect answers when they are confident in their own responses. Conversely, they are more likely to be influenced by others' opinions when uncertain about their answers. Moreover, Wang et al. (2023c) found that the probability of a correct answer decreases as the generated responses become more contradictory. Therefore, there's a high probability that the model's answer is incorrect when it frequently changes during the communication. We propose calculating the model's confidence based on the variation in responses. This aids the recipient of the information in ascertaining its reliability, thereby safeguarding the problem-solving process from the disruption of erroneous information.

In a communication with $k$ rounds, model $m_i$ generates a set of answers $\{a_i^{(1)}, \ldots, a_i^{(k)}\}$. Let $f(a_i) = \mathrm{argmax}_{a_i} \#\{a_i | a = a_i^{(j)}\}$ denote the most frequently occurring answer from model $m_i$. Consequently, we obtain the model's confidence level $\mathcal{C}_i = \frac{f(a_i)}{k}$ in the current round.

## 5 Experiments

### 5.1 Experimental Setups

**Tasks and Datasets.** In our experiments, we evaluated the performance of EoT across three com-

plex reasoning tasks: **(1) Mathematical Reasoning**: This involved six datasets, which includes GSM8K (Cobbe et al., 2021), MultiArith (Roy and Roth, 2015), SingleEQ (Koncel-Kedziorski et al., 2015), AddSub (Hosseini et al., 2014), AQuA (Ling et al., 2017), and SVAMP (Patel et al., 2021). **(2) Commonsense Reasoning**: We utilized the CommonsenseQA(CSQA; Talmor et al., 2019) and StrategyQA (Geva et al., 2021). **(3) Symbolic Reasoning**: We employed two datasets from Big-Bench (bench authors, 2023; Suzgun et al., 2023), namely Penguins in a Table (Penguins) and Date Understanding. In Appendix B, we provide a detailed description and statistics of the datasets.

**Baselines.** We compare EoT with a series of strong baselines, which include (1) Chain-of-Thought prompting (CoT; Wei et al., 2022b), (2) Complexity-based prompting (ComplexCoT; Fu et al., 2023b), (3) Self-Consistency (SC; Wang et al., 2023c), (4) Progressive Hint Prompting (PHP; Zheng et al., 2023). Specifically, CoT and ComplexCoT are prompting methods, while SC and PHP are reasoning chain ensemble methods. For simplicity in notation, we use "CoT-SC(10)" to denote the approach that employs the CoT prompt method to sample 10 reasoning chains and then utilize the SC method to select the answer.

**Implementation Details.** We accessed the GPT models through the OpenAI API. In the main experiments, we employed `GPT-3.5-Turbo-0301` (GPT-3.5) and `GPT-4-0314` (GPT-4) to evaluate the effectiveness of EoT in comparison to other strong baselines. We set the temperature at 1 during the generation. The prompts for CoT and ComplexCoT are sourced from Wei et al. (2022b) and Fu et al. (2023b). By default, we employ three `GPT-3.5-Turbo-0301` to engage in the EoT communication. We apply the majority consensus termination and confidence evaluation, selecting the majority answer as the final outcome. Taking into account the impact of temperature, we report the average performance and standard deviation across five runs. Additionally, in section 5.3, to further validate the performance of different LLMs on EoT, we incorporated the `Claude-2` model. The further implementation details are listed in Appendix C.

### 5.2 Performance of EoT

**Mathematical Reasoning.** According to the results presented in Table 1, the four communication paradigms of EoT have shown significant improve-

| Method | GSM8K | MultiArith | SingleEQ | AddSub | AQuA | SVAMP | Avg. |
|---|---|---|---|---|---|---|---|
| *Single Reasoning Chain* | | | | | | | |
| CoT | 79.12±0.50 | 97.27±0.65 | 92.80±0.27 | 86.23±0.52 | 55.12±1.03 | 79.52±0.81 | 81.67 |
| ComplexCoT | 79.32±0.65 | 95.40±0.50 | 91.34±0.33 | 84.46±0.86 | 56.46±0.59 | 77.70±0.54 | 80.78 |
| CoT (GPT-4) | **94.90** | 97.80 | 93.10 | **89.30** | **77.50** | **90.50** | 90.51 |
| *Ensemble Methods* | | | | | | | |
| CoT-SC(3) | 82.82±0.32 | 98.20±0.43 | 93.31±0.12 | 87.19±0.47 | 62.13±1.30 | 81.98±0.49 | 84.27 |
| CoT-SC(5) | 85.47±0.52 | 98.60±0.08 | 93.70±0.25 | 87.49±0.38 | 64.02±0.95 | 83.76±0.81 | 85.50 |
| CoT-SC(10) | 87.57±0.27 | 98.97±0.12 | 94.06±0.36 | 87.59±0.58 | 66.38±1.72 | 84.96±0.33 | 86.59 |
| ComplexCoT-SC(3) | 84.17±0.67 | 97.43±0.31 | 92.95±0.53 | 86.13±0.74 | 60.47±1.55 | 81.44±0.79 | 83.77 |
| ComplexCoT-SC(5) | 87.26±0.33 | 98.13±0.22 | 94.02±0.29 | 86.48±0.61 | 62.05±2.40 | 83.86±0.92 | 85.30 |
| ComplexCoT-SC(10) | 89.23±0.31 | 98.23±0.37 | 94.21±0.16 | 86.58±0.58 | 64.96±1.91 | 85.58±0.87 | 86.46 |
| PHP | 85.10 | 98.00 | 92.90 | 85.30 | 60.60 | 83.10 | 84.16 |
| *Exchange-of-Thought* | | | | | | | |
| EoT-Memory | 88.98±0.89 | 98.80±0.16 | 94.09±0.48 | 87.65±0.49 | 69.37±2.77 | 84.28±0.48 | 87.20 |
| EoT-Report | 88.61±0.83 | **99.03**±0.22 | 94.06±0.47 | 87.95±0.34 | 70.31±2.19 | 84.78±0.75 | 87.46 |
| EoT-Relay | 88.42±0.72 | 98.97±0.16 | 94.13±0.49 | 87.59±0.58 | 70.87±1.98 | 85.04±0.31 | 87.50 |
| EoT-Debate | 88.52±0.76 | 98.90±0.17 | **94.25**±**0.19** | 87.70±0.34 | 69.69±1.24 | 85.10±0.24 | 87.36 |

Table 1: Comparison of EoT performance with a series of strong baselines on mathematical reasoning tasks. Best results are highlighted in bold, while the best results among different EoT modes are underlined. The experimental results for CoT (GPT-4) and PHP were sourced from Zheng et al. (2023).

ment over both CoT and ComplexCoT in mathematical reasoning tasks. Compared to the currently strongest baseline method, PHP, the Memory, Report, Relay, and Debate paradigms have respectively increased the average performance by 3.04%, 3.30%, 3.34%, and 3.20%. EoT comprehensively outperforms CoT-SC(5), achieving performance comparable to, or even surpassing, that of CoT-SC(10). When compared to the current best LLM GPT-4, three GPT-3.5 with EoT surpassed a single GPT-4 with CoT on the MultiArith and SingleEQ datasets. This indicates that through inter-model communication and collaboration, three less capable models can compensate for their individual weaknesses and outperform more powerful model, showcasing the potential of EoT to enhance model capabilities and address inherent shortcomings by incorporating external insights.

**Commonsense Reasoning.** The comparison of EoT with CoT and CoT-SC methods on commonsense reasoning tasks is illustrated in Figures 5a and 5b. EoT significantly outperforms the CoT. Specifically, on the StrategyQA dataset, Memory, Report, Relay, and Debate respectively achieve improvements of 8.06%, 8.24%, 8.42%, and 8.67% compared to CoT. Similar significant gains are observed on the CSQA dataset. Furthermore, across both commonsense reasoning tasks, all four

paradigms outperform the CoT-SC(10) method, which samples 10 reasoning chains, demonstrating the superior performance of EoT.

**Symbolic Reasoning** Figures 5c and 5d compare the performance of EoT with CoT and CoT-SC methods on symbolic reasoning tasks. On the Penguins dataset, the Memory, Report, Relay, and Debate paradigms of EoT achieve improvements of 2.01%, 1.92%, 2.33%, and 2.05% respectively, compared to the CoT-SC(3) method which samples 3 reasoning chains. On the Date Understanding dataset, the performance gains of EoT are even more pronounced, with all four methods showing an average improvement of 2.1% over CoT-SC(10).

### 5.3 Discussions

**Communication Mode.** We propose four communication paradigms and analyze the communication volumes in section 4.1 and section 4.2. In the results illustrated in Table 1, we observe that different communication paradigms have their respective strengths. For instance, Report performs best on MultiArith and AddSub, while Debate achieves optimal performance on SingleEQ and SVAMP. This suggests that various communication paradigms are best suited for different scenarios.

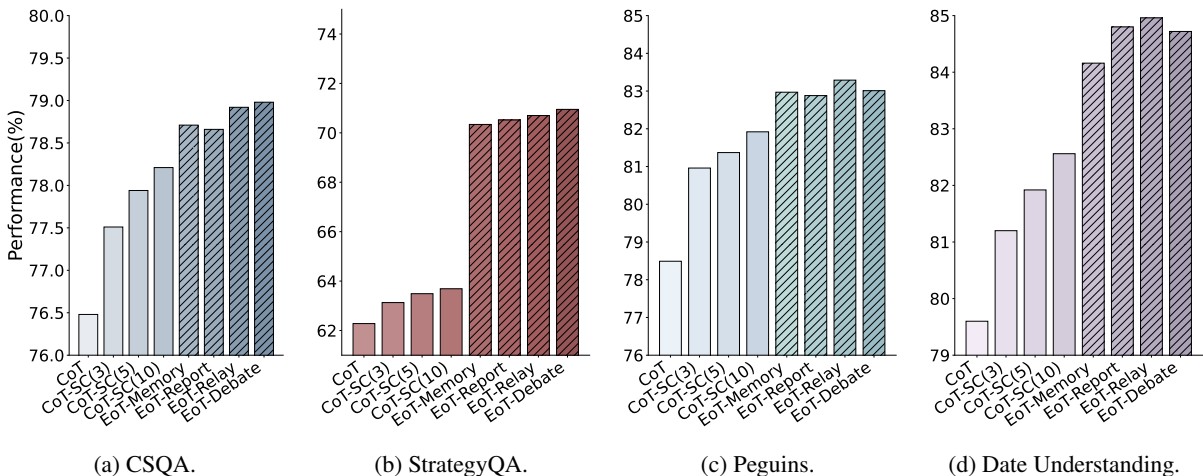

(a) CSQA.  (b) StrategyQA.  (c) Peguins.  (d) Date Understanding.

Figure 5: Comparison of EoT with CoT and CoT-SC methods in commonsense and symbolic reasoning tasks.

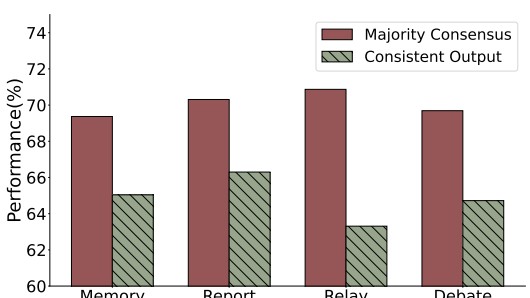

Figure 6: Comparison of consistent output termination and majority consensus termination on AQuA.

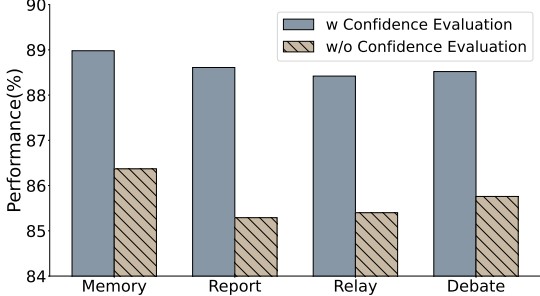

Figure 7: The impact of employing confidence evaluation on accuracy in the GSM8K dataset.

**Termination Condition.** In Figure 6, we analyze the performance of the two termination conditions we propose on the AQuA dataset. Compared to consistent output termination, majority consensus termination improved by 4.33%, 4.01%, 7.56%, and 4.97% under the Memory, Report, Relay, and Debate paradigms, respectively. Under consistent output termination, there is no mechanism for collective negotiation, and individual models are prone to premature exit due to degeneration (Su et al., 2022). Therefore, majority consensus termination is more suitable for scenarios involving multiple model communication.

**Confidence Evaluation.** We conduct ablation experiments on the GSM8K dataset for confidence evaluation. As shown in Figure 7, across four communication paradigms, confidence evaluation showed an average improvement of 2.92% compared to the baseline. The introduction of confidence evaluation enables the model to consider the other model's confidence prior (Zhang et al., 2023a) during communication, facilitating its decision to accept the other model's reasoning chains

at an earlier stage, thereby effectively mitigating the interference of incorrect reasoning chains.

**Round Analysis.** As illustrated in Figure 8, we analyze the number of communication rounds to satisfy termination condition in the SVAMP dataset. For the majority of samples, consensus on the answer can be reached within three rounds of communication. Wang et al. (2023c) obverse that answer consistency is proportional to accuracy. EoT enables models to engage in a greater number of exchanges and discussions on questions where consensus is challenging to achieve. Consequently, a minority of difficult cases necessitate communication extending beyond five rounds.

**Cost Analysis.** A potential concern is the computational expense incurred by EoT. In Figure 9, we compare the performance and computational costs of CoT-SC, ComplexCoT-SC, and EoT methods. Compared to CoT-SC(5), EoT reduces costs by 20% while enhancing performance by 3%. EoT achieves comparable performance to ComplexCoT-SC(10) at only one-seventh of its cost. Since the majority of samples conclude communication

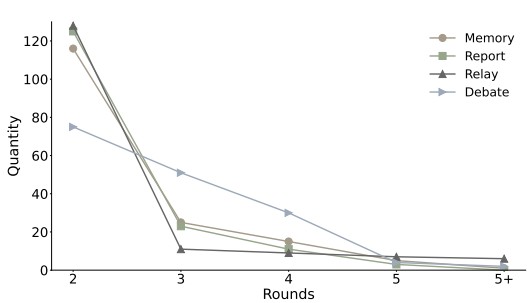

Figure 8: Number of communication rounds required to reach termination condition on SVAMP.

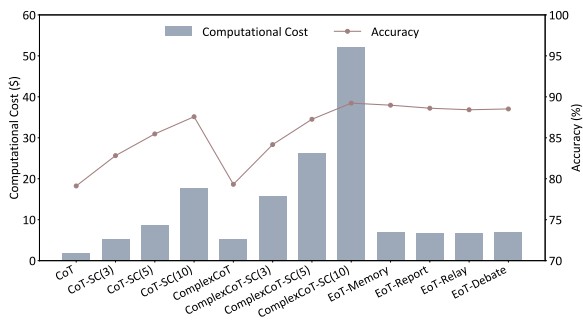

Figure 9: Performance and associated costs of different methods in the GSM8K dataset.

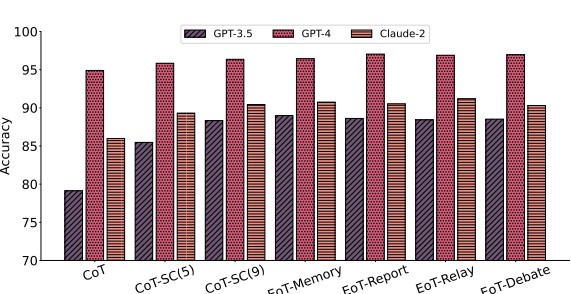

Figure 10: Comparison of EoT with CoT and CoT-SC methods using different LLMs as backbones on GSM8K.

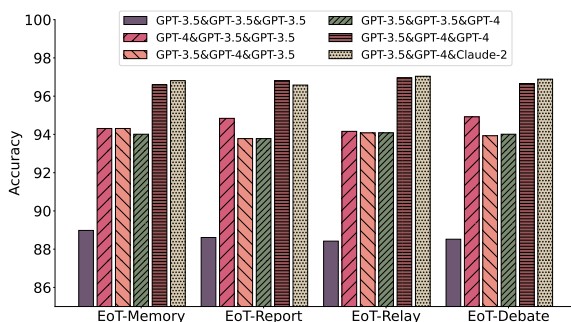

Figure 11: Effect of different node positions for LLMs on accuracy in the GSM8K Dataset.

within three rounds, EoT does not impose a significant computational burden. By facilitating the exchange of external insights between models, EoT effectively enhances model performance, demonstrating a cost-effective advantage.

**Model Applicability.** In Figure 10, we analyze the performance of EoT when applied to different LLMs. EoT, compared to CoT-SC(5), shows performance improvements of 3.2% on GPT-3.5, 1.0% on GPT-4, and 1.4% on Claude-2, indicating that EoT is adaptable to various LLMs and effectively boosts performance across multiple LLMs.

**Position Analysis.** In Figure 11, we investigate the impact of different LLMs occupying different node positions on performance. Notably, positioning the more powerful GPT-4 as the central node in the Report paradigm yields a performance increase of over 1% compared to when GPT-4 serves as a non-central node. In the Debate paradigm, GPT-4 as a parent node outperforms GPT-4 as a child node by 0.9%. The location of GPT-4 has a negligible effect on the decentralized Relay and Memory paradigms. Additionally, a configuration with two GPT-4 models and one GPT-3.5 significantly outperforms one with two GPT-3.5 models

and one GPT-4, underscoring that incorporating more superior models can further enhance EoT's performance. The combination of GPT-3.5, GPT-4, and Claude-2 achieves performance close to or exceeding that of two GPT-4 with one GPT-3.5, suggesting that model diversity can effectively boost EoT's effectiveness, aligning with the ensemble theory (Kuncheva and Whitaker, 2003) that diversity among models can improve performance.

## 6 Conclusion

We introduce Exchange-of-Thought (EoT), a novel framework that enriches models with external insights through cross-model communication. We develop four communication paradigms and conduct a thorough analysis of the communication volume and information propagation speed. To safeguard against the disruption of incorrect reasoning processes, we design a confidence evaluation mechanism. Experiment on mathematical, commonsense, and symbolic reasoning tasks demonstrate that EoT surpasses a series of strong baselines while also offering a cost advantage. Further analysis reveals that EoT is adaptable to various models, and the participation of a more diverse range of models can further enhance the performance of EoT.

## Ethics Statement

The EoT method presented in this paper does not require the collection or utilization of any personal information. The prompts we have designed and employed are free from personal data and avoid language that discriminates against individuals or groups. We have conducted a comprehensive research of the licenses for the datasets used in this paper, as detailed in Appendix B, and have ensured that our work complies with all the licensing requirements of these datasets.

## Acknowledgement

This work was supported by the National Key Research and Development Program of China (No.2022CSJGG0801), National Natural Science Foundation of China (No.62022027). We extend our sincerest gratitude to the reviewers for their insightful comments and suggestions, which have been instrumental in enhancing the quality of this manuscript.

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

## A Limitations and Broader Impacts

Considering the limitations in communication and analytical capabilities of open-source models (Fu et al., 2023a), as well as their computational resource requirements (Touvron et al., 2023b; Chowdhery et al., 2022), we have not included them in our experiment at this point. Nonetheless, we believe that open-source models with proficient comprehension and communication abilities could achieve, or even surpass, the performance of commercial models (OpenAI, 2023; Ouyang et al., 2022) through mutual exchange of insights.

A fundamental aspect of model communication is the handling of long text. Given the current context limitations of the models, we did not include a larger number of models in the communication process. Recent works (Liu et al., 2023; Xiao et al., 2023; Wang et al., 2023b; Tworkowski et al., 2023; Chen et al., 2023; Ratner et al., 2023, *inter alia*) that equip large language models with the ability to process extended texts lay the groundwork for broader model communication. During our experiments, we observed that model communication can achieve comparable performance with lower computational resources, paving the way for more sustainable development of AI (Van Wynsberghe, 2021; Wu et al., 2022).

Furthermore, the concept of AI learning from each other to foster collective improvement is a focal point of current research (Bai et al., 2022b; Lee et al., 2023; Ponnusamy et al., 2022). Our vision and hope are to create a collective intelligence among LLMs (Ha and Tang, 2022).

| Dataset | Reasoning Task | Ans Type | # Prompt | # Test | License |
|---|---|---|---|---|---|
| GSM8K (Cobbe et al., 2021) | Mathematical | Number | 8 | 1,319 | MIT License |
| MultiArith (Roy and Roth, 2015) | Mathematical | Number | 8 | 600 | Unspecified |
| SingleEq (Koncel-Kedziorski et al., 2016) | Mathematical | Number | 8 | 508 | Unspecified |
| AddSub (Hosseini et al., 2014) | Mathematical | Number | 8 | 395 | Unspecified |
| AQUA (Ling et al., 2017) | Mathematical | Multi-choice | 4 | 254 | Apache-2.0 |
| SVAMP (Patel et al., 2021) | Mathematical | Number | 8 | 1,000 | MIT License |
| CommonsenseQA (Talmor et al., 2019) | Commonsense | Multi-choice | 7 | 1,221 | Unspecified |
| StrategyQA (Geva et al., 2021) | Commonsense | T/F | 6 | 2,290 | MIT license |
| Date Understanding (Suzgun et al., 2023) | Symbolic | Multi-choice | 3 | 369 | MIT license |
| Penguins in a Table (Suzgun et al., 2023) | Symbolic | Multi-choice | 3 | 146 | MIT license |

Table 2: Detailed statistics of the datasets utilized in our experiment. Ans Type indicates the form of the answer. # Prompt represent the count of chain-of-thought exemplars employed as few-shot prompts for each task. # Test indicates the quantity of samples contained within each dataset.

## B    Datasets and Evaluation Metrics

**Datasets**    In Table 2, we meticulously detail the specifics and statistics of each dataset employed in our experiments. This includes the data source, task type, answer type, the number of prompt samples used, the total number of test samples, as well as the licenses pertaining to each dataset.

**Evaluation Metrics**    Accuracy is used as the metric for evaluation in our study. For datasets where the answer is numerical, we employ regular expressions to extract the number following the phrase "the answer is" and perform a numerical comparison with the provided answer. For datasets with multiple-choice and true/false questions, accuracy is calculated by checking if the option extracted from the output matches the correct answer.

In the main experiment, all test samples are used for evaluation. In the analysis part, due to rate limits and cost considerations, we set an upper limit on the sample size. Consequently, a maximum of 1,000 samples are utilized for each run.

## C    Implementation Details

**Confidence Evaluation.**    Considering that confidence evaluation requires historical answers for reference, we begin incorporating the confidence information into the prompts from the second round of communication. Specifically, after calculating $C_i$ using the method described in Section 4.4, we preface the solution with "$\mathcal{M}_i$'s confidence in this solution is $\mathcal{C}_i$", where $\mathcal{M}_i$ is the character name.

**Termination Condition.**    For the consistent output termination condition, a minimum of two rounds of communication is necessary, as it requires the model's answer from the previous round. Given that only three models are involved in the EoT communication, the exit of a single model

reduces the interaction to a dialogue between the remaining two, potentially impeding their communication. Therefore, if a single model exits, we terminate the communication and select the exiting model's answer as the final result.

In the case of majority consensus termination, if the answers from all three models align in the first round, we deem further communication unnecessary and end the exchange. Given that only three models are involved in the communication, an exit based on two models holding the same incorrect answer could lead to an inaccurate final result. Therefore, during the initial five rounds, we require a unanimous agreement among all models before ceasing communication. If a consensus is not reached after five rounds, the majority answer will be adopted as the final outcome.

**Computation Cost.**    Computational costs are calculated based on OpenAI's official pricing for `GPT-3.5-Turbo-0301`, which is computed as Input Tokens $\times 0.0015/1000$ + Output Tokens $\times 0.002/1000$.

## D    EoT Prompts

During the EoT communication process, we assign different roles to the models. Table 3 displays the prompts for each role, wherein we have models A, B, and C take on the personas of Kitty, Ben, and Peter, three high school students, to facilitate the communication. The specific prompts for different datasets can be found in our Github repository.

## E    Case Studies

To enhance our understanding of the four communication paradigms, we have conducted case studies for each paradigm. The processes of each communication paradigm are illustrated in Tables 4, 5, 6, and 7 respectively.

## CHARACTER PROMPTS

 **Kitty**: "You are Kitty, a high school student admired for your attentiveness and detail-oriented nature. Your friends often rely on you to catch details they might have missed in their work. Your task is to carefully analyze the presented math problem, apply your attentive skills, and piece together a detailed solution. Afterward, you'll have the opportunity to review the solutions provided by Ben and Peter, offering insights and suggestions. Your careful revisions will help all of you to enhance your understanding and arrive at the most accurate solutions possible."

 **Ben**: "You are Ben, a high school student with a track record of excellent grades, particularly in mathematics. Your friends admire your diligence and often seek your guidance in their studies. Your role is to scrutinize the problem at hand with your usual attention to detail, drawing from your vast knowledge of math principles. After considering your friends' approaches, carefully construct your answer, ensuring to clarify each step of your process. Your clear and logical explanations are valuable, as they will serve as a benchmark for your friends to compare and refine their own solutions."

 **Peter**: "You are Peter, a high school student recognized for your unique problem-solving abilities. Your peers often turn to you for assistance when they encounter challenging tasks, as they appreciate your knack for devising creative solutions. Today, your challenge is to dissect the given math problem, leveraging your unique problem-solving strategies. Once you've crafted your solution, share it with your friends, Ben and Kitty, so they can see a different perspective. Your innovative approach will not only provide an answer but also inspire Ben and Kitty to think outside the box and possibly revise their own solutions."

## COMMUNICATION PROMPTS

Please consider the example provided and think it step by step.
Question: {}
Here is a solution process from your friend:
Solution: {}
Your friend's confidence in this solution is: {}
Based on your friend's solution, carefully re-examine your previous answer. If your friend's confidence level is below 0.5, it suggests a high probability that the solution might be incorrect. Remember, solutions with high confidence can also be wrong. Utilize your talent and critical thinking to provide a new step-by-step solution process.

Table 3: Examples of system prompts used for role-playing and communication prompts in EoT communication.

**Q**: Carla is downloading a 200 GB file. Normally she can download 2 GB/minute, but 40% of the way through the download, Windows forces a restart to install updates, which takes 20 minutes. Then Carla has to restart the download from the beginning. How load does it take to download the file?

**Kitty's Solution**: Carla was able to download for 40% of the time before the restart. 40% of 200 GB is 80 GB. Since she can download 2 GB/minute, it would take her 80/2 = 40 minutes to download 80 GB. After the forced restart, she has to restart the download from the beginning, so she would need another 200/2 = 100 minutes to complete the download. Plus the 20 minutes for the restart, the total time would be 40 + 100 + 20 = 160 minutes. So the answer is 160 minutes.

**Ben's Solution**: Carla needs to download a 200 GB file. Normally, she can download 2 GB/minute, which means it would take her 200 / 2 = 100 minutes to download the file. However, 40% of the way through the download, Windows forces a restart for 20 minutes, which means Carla has downloaded 200 * 0.4 = 80 GB before the restart. After the restart, she needs to start the download again from the beginning, so the total time it takes her would be 100 + (200 / 2) = 200 minutes. Therefore, it takes Carla 200 minutes to download the file. So the answer is 200 minutes.

**Peter's Response**: Upon re-evaluating the problem and reviewing Kitty and Ben's solutions, it's clear their results deviated due to a misunderstanding of the problem conditions - primarily, how much of the downloaded file was retained after Window's forced restart. To clarify, here is the new step-by-step solution:
1. Start with Carla's download rate: she can download 2 GB/minute.
2. Therefore, in theory, if uninterrupted, Carla would be able to download the 200 GB file in 200 GB / 2 GB/minute = 100 minutes.
3. However, 40% of the way through the download, Windows forced a restart, so Carla had already downloaded 0.40 x 200 GB = 80 GB at that time. This took 80 GB / 2 GB/minute = 40 minutes.
4. The problem statement clarifies that after the restart, Carla has to start again from the beginning. This means the 80 GB downloaded before the forced restart was wasted.
5. Consequently, after the restart, Carla still needs to download the whole 200 GB file, which, at her rate, will take her another 200 GB / 2 GB/minute = 100 minutes.
6. We cannot forget the time for the Windows restart: it took an additional 20 minutes.
7. Consequently, the total time to download the file, taking into account the restart, is the sum of the time for the initial download attempt, the time for the restart, and the time for the full download post-restart. It results in 40 minutes (initial incomplete download) + 20 minutes (Windows restart) + 100 minutes (full download post-restart) = 160 minutes.
From these calculations, it appears that Kitty's solution is correct. It takes Carla 160 minutes to download the file, given the interruption due to Windows update that forced her to restart the download from the beginning. So the answer is 160 minutes.

Table 4: An illustrative example of the Memory paradigm of EoT in the GSM8K dataset.

**Q**: Carla is downloading a 200 GB file. Normally she can download 2 GB/minute, but 40% of the way through the download, Windows forces a restart to install updates, which takes 20 minutes. Then Carla has to restart the download from the beginning. How load does it take to download the file?

**Ben's Solution**: Carla needs to download a 200 GB file. Normally, she can download 2 GB/minute, which means it would take her 200 / 2 = 100 minutes to download the file. However, 40% of the way through the download, Windows forces a restart for 20 minutes, which means Carla has downloaded 200 * 0.4 = 80 GB before the restart. After the restart, she needs to start the download again from the beginning, so the total time it takes her would be 100 + (200 / 2) = 200 minutes. Therefore, it takes Carla 200 minutes to download the file. So the answer is 200 minutes.

**Peter's Solution**: Carla can download 2 GB in 1 minute, so to download 200 GB, it will take her 200/2 = 100 minutes if she didn't have to restart. However, she was 40% of the way through the download when Windows forced the restart, so she had already downloaded 0.4 x 200 = 80 GB. The restart took 20 minutes, so she lost 20 x 2 = 40 GB of progress. After the restart, she had to start again from 0 GB. So in total, she had to download 200 - 80 - 40 = 80 GB. At her regular download speed of 2 GB/minute, it will take her 80/2 = 40 minutes to download the remaining 80 GB. Adding the 20 minutes of the Windows update, the total time it takes Carla to download the file is 20 + 40 = 60 minutes. So the answer is 60 minutes.

**Kitty's Solution**: Carla was able to download for 40% of the time before the restart. 40% of 200 GB is 80 GB. Since she can download 2 GB/minute, it would take her 80/2 = 40 minutes to download 80 GB. After the forced restart, she has to restart the download from the beginning, so she would need another 200/2 = 100 minutes to complete the download. Plus the 20 minutes for the restart, the total time would be 40 + 100 + 20 = 160 minutes. So the answer is 160 minutes.

**Kitty's Response**: Analyzing both Ben's solution and Peter's solution, Peter's conclusion is incorrect. He misunderstood the situation. After the Windows restart, the download does not continue from where it stopped. It starts from the beginning, which means the download progress does not reduce what is left to download. Revisiting Ben's solution, he almost got everything right except the final calculation step whereas he added an extra 100 minutes. The correct calculation should be:
Before forced restart: 80 GB (40% of the total file) in 40 minutes (80/2 = 40)
After forced restart: 200 GB total file in 100 minutes (200/2 = 100)
Windows restart: 20 minutes
Total download time = Before restart time + After restart time + Windows restart time
Total download time = 40 minutes + 100 minutes + 20 minutes = 160 minutes
Ben's setup was correct, but he added an extra 100 minutes due to an arithmetic mistake. The download time in his calculations should total to 160 minutes, not 200 minutes. So, effectively upon examining both scenarios by Peter and Ben, when taking all aspects into account and considering the restart, the total time it should take Carla to download the file is 160 minutes in total. So the answer is 160 minutes.

Table 5: An illustrative example of the Report paradigm of EoT in the GSM8K dataset.

**Ben's Solution**: Carla needs to download a 200 GB file. Normally, she can download 2 GB/minute, which means it would take her 200 / 2 = 100 minutes to download the file. However, 40% of the way through the download, Windows forces a restart for 20 minutes, which means Carla has downloaded 200 * 0.4 = 80 GB before the restart. After the restart, she needs to start the download again from the beginning, so the total time it takes her would be 100 + (200 / 2) = 200 minutes. Therefore, it takes Carla 200 minutes to download the file. So the answer is 200 minutes.

**Kitty's Response**: After reviewing Ben's solution, it appears there seems to be a mathematical error in the final part of his solution. While he correctly calculates that Carla has downloaded 80 GB of the file before the restart, he fails to account for the time Carla already spent downloading the 80GB in his final total. Furthermore, he miscalculated the time it takes Carla to download the file after the restart. Now, let's correct the calculation. Before the restart, Carla downloaded 40% of the file, which is 40/100 * 200 = 80 GB. Given she downloads 2 GB per minute, this would have taken her 80 / 2 = 40 minutes. Unfortunately, the computer restarts, wasting 20 minutes of her time. After the restart, Carla has to download the entire file again. With her download speed, this will take another 200 / 2 = 100 minutes. Therefore, the total time Carla spends downloading the file is the time before the restart + the time wasted on the restart + the time after the restart, which is 40 minutes + 20 minutes + 100 minutes = 160 minutes. So, the final and correct answer should indeed be 160 minutes.

**Peter's Solution**: Carla can download 2 GB in 1 minute, so to download 200 GB, it will take her 200/2 = 100 minutes if she didn't have to restart. However, she was 40% of the way through the download when Windows forced the restart, so she had already downloaded 0.4 x 200 = 80 GB. The restart took 20 minutes, so she lost 20 x 2 = 40 GB of progress. After the restart, she had to start again from 0 GB. So in total, she had to download 200 - 80 - 40 = 80 GB. At her regular download speed of 2 GB/minute, it will take her 80/2 = 40 minutes to download the remaining 80 GB. Adding the 20 minutes of the Windows update, the total time it takes Carla to download the file is 20 + 40 = 60 minutes. So the answer is 60 minutes.

**Ben's Response**: Upon reviewing both solutions, it seems there might have been some misunderstandings. As per the problem, Carla can download at 2 GB/minute, and she needs to download a 200 GB file. By this rate, it would take her 200 GB / 2 GB per minute = 100 minutes without any interruptions. However, 40% of the way through the download, Windows force restart happened which lasted for 20 minutes. By that time, she downloaded 40% of her file which is equal 200 GB * 0.4 = 80 GB. The restart not only consumed 20 minutes but also voided her previous download progress, which means after the updates she had to start downloading all over again. So, the total time taken by Carla to download the 200 GB file would be calculated as: Regular download time (200 GB at 2 GB/minute) + Time spent in updates = (200 / 2) + 20 = 100 + 20 = 120 minutes. So the correct total time it takes Carla to download the file is 120 minutes.

Table 6: An illustrative example of the Relay paradigm of EoT in the GSM8K dataset. Kitty firmly maintains her correct answer, while Ben's response begins to waver.

**Q**: Carla is downloading a 200 GB file. Normally she can download 2 GB/minute, but 40% of the way through the download, Windows forces a restart to install updates, which takes 20 minutes. Then Carla has to restart the download from the beginning. How load does it take to download the file?

**Peter's Response**: Analyzing both solutions, it appears that there is a slight misunderstanding in Ben's solution. He has missed considering the download progress lost due to the forced restart of Windows. Let's modify and re-explain the solution in a way that takes this into account:

Step 1: Calculate the time for her initial progress. Carla was able to download 40% of the file before the interruption. That is, she downloaded 0.4 * 200GB = 80GB. With her speed of 2GB/minute, she used 80/2 = 40 minutes on this initial download.

Step 2: Consider the time lost due to the forced restart. The Windows forced update lasted for 20 minutes, during which time she made no progress on her download.

Step 3: Calculate the remaining time to finish the download. After the forced restart, Carla had to re-download the whole file. It takes her 200GB / 2GB per minute = 100 minutes at her download speed to download the whole file.

Step 4: Calculate the total time. The total time it takes Carla to download the whole file, considering the interruption, is the sum of the time it took for her initial progress, the time lost due to the forced restart, and the time it took for her to finish the download after the restart. This is 40 minutes + 20 minutes + 100 minutes = 160 minutes.

Therefore, the correct answer is 160 minutes. This improved solution accounts for the hiccup caused by the Windows update which Ben did not factor into his original problem-solving strategy.

**Ben's Response**: Reviewing the problem again, I can see that there are errors in both my solution and my friend Peter's solution. Here is the corrected step-by-step solution:

1. Carla downloaded 40% of the 200 GB file before the system restart. This equals 40/100 * 200 = 80 GB.

2. At a rate of 2 GB/minute, it must have taken her 80 / 2 = 40 minutes to download that 80 GB.

3. Then, the Windows update interrupted the download for 20 minutes.

4. After the restart, she had to start again from the beginning, so she needs again 200 GB/2 GB per minute = 100 minutes to download the full file.

5. If we add up these times, we get the total time Carla spent on the download: 40 + 20 + 100 = 160 minutes.

So, it would take Carla 160 minutes to download the file, correcting both my previous response and Peter's calculations. This solution process allows for the clear understanding of the time taken to download the file considering both the time before and after the interruption. So the answer is 160 minutes.

Table 7: An illustrative example of the debate paradigm of EoT in the GSM8K dataset. Through mutual discussion, Ben and Peter identify the flaws in their respective solutions and arrive at the correct answer.