# OpenReview forum: "Exchange-of-Thought: Enhancing Large Language Model Capabilities through Cross-Model Communication"
_EMNLP/2023/Conference — EMNLP 2023 Main_

### Official Review · Reviewer_u6cP · 2023-08-02

**Typos Grammar Style And Presentation Improvements:** Line 103
**Soundness:** 4

**Excitement:**

4: Strong: This paper deepens the understanding of some phenomenon or lowers the barriers to an existing research direction.

**Missing References:**

- For communication in multi-agent system and a further exploration of communciation modes: Parsons, S., McBurney, P. (2003). Argumentation-Based Communication between Agents. In: Huget, MP. (eds) Communication in Multiagent Systems. Lecture Notes in Computer Science(), vol 2650. Springer, Berlin, Heidelberg. https://doi.org/10.1007/978-3-540-44972-0_8

**Paper Topic And Main Contributions:**

This paper presents a novel method, called "Exchange-of-Thought," to enhance the reasoning capabilities of LLMs (Language Language Models) in mathematical and common-sense reasoning tasks. It si based on the Chain-of-Thought concept, generates intermediate reasoning steps for addressing complex problems. By introducing four communication modes, the paper argues that it enhances the quality and reliability of the answers with respect to the state of the art. Experimental validation shows that this interactive communication approach outperforms other methods.

**Questions For The Authors:**

- Is this paper considering communication using the concepts from multi-agent systems?

**Reasons To Accept:**

- The paper addresses a relevant issue: enhancing the reasoning capabilities of LLMs.
- The interactive and multi-agent approach is an interesting and novel proposal for tackling this challenge.
- The results of the experiments are significant and improve the state of the art.

**Reasons To Reject:**

- Lack of reference explaining communication in this context.
- The paper introduces four communication modes (debate, report, relay, and memory) without sufficient support from literature, despite existing relevant work in argumentation theory. Section 4.2 provides inadequate details and lacks illustrative examples.
- Figure 3 is challenging to understand. The workflow and captions are unclear, and the representation of communication modes on the left side is confusing.
- Figure 4's tabular representation of node agent interactions is not intuitive.

**Reproducibility:**

3: Could reproduce the results with some difficulty. The settings of parameters are underspecified or subjectively determined; the training/evaluation data are not widely available.

**Reviewer Confidence:**

2: Willing to defend my evaluation, but it is fairly likely that I missed some details, didn't understand some central points, or can't be sure about the novelty of the work.

---

> ### Author Rebuttal · Authors · 2023-08-28
>
> We are deeply grateful for the invaluable insights you provided regarding communication modes in relation to our work. We have thoroughly reviewed the literature you provided and have gained significant insights from it. The paper's approach to argumentation-based communication, requiring agents to not only send information but also to provide appropriate rationales, is particularly enlightening. From the perspective of the paper, our work could be categorized under inquiry dialogues, where agents cooperate to answer given questions, utilizing English as the shared language for communication between models. The argumentation theory presented in the paper is extremely insightful, and we have added an introduction to this aspect and cited your provided literature accordingly. We sincerely appreciate your recommendations and references, and we plan to further explore the application of argumentation theory in the communication among large language models in our future work.
>
> Our proposed modes of communication—debate, report, relay, and memory—are inspired by network topologies[1]: Tree Topology, Star Topology, Ring Topology, and Bus Topology. For further details on the correspondence between communication modes and network topologies, please refer to our response to Reviewer HLJV. We sincerely apologize for any confusion that Figures 3 and 4 may have caused. In Figure 4, the horizontal axis represents the sources from which a model is allowed to receive information, while the vertical axis indicates the permissible targets for sending information. To clarify these aspects, we've revamped the layout of these figures, aiming to provide clearer insights into the workflow of the different communication modes.
>
> To better illustrate the workflow of different communication modes, we present an analogy wherein agents take on the roles of Kitty, Alice, and Ben, representing nodes A, B, and C, respectively. Initially, each agent contemplates the problem independently, formulating their answers and rationales.
> - In the Debate mode, Kitty acts as a facilitator encouraging students Alice and Ben to discuss, but not responding to them. Hence, Alice and Ben don't receive feedback from Kitty.
> - In the Report mode, Kitty assumes the role of a strict instructor who doesn't allow Alice and Ben to communicate with each other, directing their interactions solely towards her.
> - In the Relay mode, all agents, Kitty, Alice, and Ben, interact as equals, forming a closed loop with communication flowing clockwise.
> - In the Memory mode, information from all agents, Kitty, Alice, and Ben, is logged, allowing every agent access to the previous round's input from all other agents.
>
> Let's use $Kitty_i$, $Alice_i$, and $Ben_i$ to represent the answers and rationales of agents Kitty, Alice, and Ben at the $i^{th}$ round, respectively. The notation $(Alice_i, Ben_i)$ indicates the aggregation of Alice's and Ben's information at the $i^{th}$ round. An arrow $(Alice_i, Ben_i) \rightarrow Kitty_{i+1}$ denotes that the combined information from Alice and Ben at round $i$ is sent to Kitty for generating her answers and rationales for the next round, $i+1$. By default, each agent has access to their own information from the previous round. The flow of information for each mode is as follows:
>
> - Debate: $ (Alice_i, Ben_i) \rightarrow Kitty_{i+1} $, $ Ben_i \rightarrow Alice_{i+1} $, $ Alice_i \rightarrow Ben_{i+1} $
>
> - Report: $ (Alice_i, Ben_i) \rightarrow Kitty_{i+1} $, $ Kitty_i \rightarrow Alice_{i+1} $, $ Kitty_i \rightarrow Ben_{i+1} $
>
> - Relay: $ Alice_i \rightarrow Kitty_{i+1} $, $ Ben_i \rightarrow Alice_{i+1} $, $ Kitty_i \rightarrow Ben_{i+1} $
>
> - Memory: $ (Alice_i, Ben_i) \rightarrow Kitty_{i+1} $, $ (Kitty_i, Ben_i) \rightarrow Alice_{i+1} $, $ (Kitty_i, Alice_i) \rightarrow Ben_{i+1} $
>
> Please allow us to clarify the specific details of the 'Consistent Output Termination' and 'Majority Consensus Termination' conditions mentioned in Section 4.2.
>
> - **Consistent Output Termination:** This condition serves as a self-exit mechanism for agents. An agent will leave the conversation if its answer remains the same for two consecutive turns. The dialogue system terminates when the number of active agents falls below two or reaches the preset maximum number of dialogue rounds.
> - **Majority Consensus Termination:** The self-exit mechanism in 'Consistent Output Termination' could gradually reduce the number of agents, affecting the system's ability to generate and exchange information. Considering that real-life dialogues often end when a consensus is reached, and given that large language models have been shown to reach a consensus through interaction[2], this condition terminates the dialogue when the majority of agents agree on an answer.
>
> To better illustrate how these termination conditions operate, we created a Toy Example:
> - Consistent Output Termination:
>   - Turn 1: Kitty's Answer=10, Alice's Answer=12, Ben's Answer=6
>   - Turn 2: Kitty's Answer=8, Alice's Answer=4, Ben's Answer=6 [Ben exits as his answer is the same in turn 1 and turn 2.]
>   - Turn 3: Kitty's Answer=20, Alice's Answer=4 [Alice exits as her answer is the same in turn 2 and turn 3. Dialogue terminates as fewer than two agents remain.]
> - Majority Consensus Termination:
>   - Turn 1: Kitty's Answer=10, Alice's Answer=12, Ben's Answer=6
>   - Turn 2: Kitty's Answer=8, Alice's Answer=4, Ben's Answer=6
>   - Turn 3: Kitty's Answer=8, Alice's Answer=12, Ben's Answer=6
>   - Turn 4: Kitty's Answer=8, Alice's Answer=8, Ben's Answer=6 [Dialogue terminates as Kitty and Alice agree on the answer this round.]
>
> The final system output will be determined through a majority vote based on the last answers from each agent before they exit. We will include these details and examples in our paper. We sincerely appreciate the invaluable references and suggestions you've provided. Your insights into argumentation theory and multi-agent systems have been extremely enlightening and have significantly aided us in improving our work. We look forward to further exploring these concepts in our future research and thank you once again for your invaluable guidance and suggestions!
>
> References:
>
> [1] Bisht N, Singh S. Analytical study of different network topologies[J]. International Research Journal of Engineering and Technology (IRJET), 2015, 2(01): 88-90.
>
> [2] Du Y, Li S, Torralba A, et al. Improving Factuality and Reasoning in Language Models through Multiagent Debate[J]. arXiv preprint arXiv:2305.14325, 2023.

---

### Official Review · Reviewer_HLJV · 2023-08-03

**Typos Grammar Style And Presentation Improvements:** 1. In line 203, there seems Wang et a…
**Soundness:** 3

**Excitement:**

3: Ambivalent: It has merits (e.g., it reports state-of-the-art results, the idea is nice), but there are key weaknesses (e.g., it describes incremental work), and it can significantly benefit from another round of revision. However, I won't object to accepting it if my co-reviewers champion it.

**Paper Topic And Main Contributions:**

This paper proposes "exchange of thought" which contains four ways of communication, encouraging multiple LLMs to help each other with problem-solving.

Main contributions:
1. thorough experiments analyzing the communication volume and performance under four communication modes.
2. significant performance improvement over mathematical and commonsense tasks.

**Questions For The Authors:**

1. What's the difference if multiple models do not share the same backbone(eg. GPT 3.5)?
2. Could you add more analysis about the relationship between the four modes and the explanation of their performance differences?


**Reasons To Accept:**

1. Interesting cooperation modes between multiple LLMs. Thorough analysis.
2. Results of the experiments look great.

**Reasons To Reject:**

1. Four modes of communication are not utilized jointly, which makes this paper look like a work of multiple trivial ideas.
2. Careless writing.

**Reproducibility:**

4: Could mostly reproduce the results, but there may be some variation because of sample variance or minor variations in their interpretation of the protocol or method.

**Reviewer Confidence:**

4: Quite sure. I tried to check the important points carefully. It's unlikely, though conceivable, that I missed something that should affect my ratings.

---

> ### Author Rebuttal · Authors · 2023-08-28
>
> Thank you for your constructive feedback. We genuinely appreciate your recognition of our work's thorough analysis and the interesting communication modes we have proposed for multiple LLMs.
>
> Inspired by network topologies[1], we introduced four distinct communication modes:
>
> 1. The Report mode is influenced by the Star Topology. In this mode, Node A acts as a central node, aggregating information from Nodes B and C. While this mode offers efficient data integration and ease of sending, its overall performance heavily depends on Node A's information processing and analytical abilities. As the number of nodes grows, this might surpass the central node's processing capacity.
> 2. The Relay mode, inspired by Ring Topology, connects nodes in a closed loop. We also specified that information should be sent in a clockwise direction. This mode benefits from reduced communication overhead, as mentioned in Section 4.1, and does away with a central node dependency. It's easily scalable but comes with the drawback of slower information propagation.
> 3. The Memory mode is modeled after the Bus Topology, with the logbook acting as the communication bus. This mode ensures high information utilization with all nodes having equal access to the information shared by others. However, it comes with higher communication costs, and the overhead for each node increases with the number of nodes.
> 4. The Debate mode is a variation of the Tree Topology. While we allow leaf nodes to communicate with each other, communication between leaf nodes and their parent is unidirectional. This ensures that parent nodes can aggregate information from child nodes. This mode offers flexibility in leveraging both the communication abilities of child nodes and the aggregation capabilities of parent nodes.
>
> Just as different network topologies have their strengths and weaknesses, so do these communication modes. Our experiments in Table 1 also indicate that no single communication mode is universally optimal. Hence, analyzing different communication modes to identify their appropriate scenarios is essential.
>
> Regarding the writing mistakes in our manuscript, we sincerely apologize for any confusion caused during your review. We have undertaken a comprehensive revision of the related works and experimental sections. Furthermore, we have referenced the appendix in the main body, revised the layout of Figure 3, and corrected errors in Figure 6. We deeply appreciate your attention to these details and once again express our regret for the oversight.
>
> > R1: What's the difference if multiple models do not share the same backbone(eg. GPT 3.5)?
>
> From an ensemble perspective in machine learning, model diversity can significantly boost ensemble performance. In light of suggestions from Reviewer UcTV, we have also incorporated computation cost into our analysis, calculated based on the number of input and output tokens as per OpenAI and Anthropic's pricing. In this experiment, we utilized three different models: gpt-3.5-turbo-0301 (GPT-3.5), gpt-4-0314 (GPT-4), and claude-2 (Claude-2). We explored the impact of various combinations, such as GPT-4, GPT-3.5, and GPT-3.5, where the order corresponds to Kitty, Alice, and Ben, respectively. The results reveal that backbone diversity positively impacts performance. For instance, in the EoT-Relay mode, a combination of GPT-4, GPT-3.5, and Claude-2 achieved an accuracy of 97.04%, surpassing a trio of GPT-4 backbones while requiring only about half the computational cost (86.33 compared to 155.02). EoT methods also show an improvement over Majority Voting. Due to time constraints, our study was limited to the GSM8K dataset, but we plan to extend our analysis to more datasets and diverse backbone combinations in future work!
>
> | Model                    | Method        | Accuracy | Computational Cost |
> | ------------------------ | ------------- | -------- | ------------------ |
> | GPT-3.5                  | CoT           | 79.53    | 1.76               |
> | GPT-3.5                  | CoT-SC(5)     | 85.90    | 8.78               |
> | GPT-3.5                  | CoT-SC(9)     | 88.32    | 15.84              |
> | GPT-3.5                  | EoT-Debate    | 89.54    | 5.97               |
> | GPT-3.5                  | EoT-Report    | 89.69    | 6.01               |
> | GPT-3.5                  | EoT-Relay     | 89.70    | 5.92               |
> | GPT-3.5                  | EoT-Memory    | 90.06    | 6.02               |
> |                          |               |          |                    |
> | GPT-4                    | CoT           | 94.90    | 44.49              |
> | GPT-4                    | CoT-SC(5)     | 95.84    | 221.71             |
> | GPT-4                    | CoT-SC(9)     | 96.36    | 400.49             |
> | GPT-4                    | EoT-Debate    | 96.96    | 154.73             |
> | GPT-4                    | EoT-Report    | 97.04    | 154.09             |
> | GPT-4                    | EoT-Relay     | 96.89    | 155.02             |
> | GPT-4                    | EoT-Memory    | 96.43    | 158.43             |
> |                          |               |          |                    |
> | Claude-2                 | CoT           | 85.98    | 20.29              |
> | Claude-2                 | CoT-SC(5)     | 89.31    | 102.30             |
> | Claude-2                 | CoT-SC(9)     | 90.43    | 186.69             |
> | Claude-2                 | EoT-Debate    | 90.30    | 71.16              |
> | Claude-2                 | EoT-Report    | 90.52    | 71.30              |
> | Claude-2                 | EoT-Relay     | 91.20    | 70.85              |
> | Claude-2                 | EoT-Memory    | 90.75    | 72.16              |
> |                          |               |          |                    |
> | GPT-4、GPT-3.5、GPT-3.5  | Majority Vote | 90.90    | 48.02              |
> | GPT-4、GPT-3.5、GPT-3.5  | EoT-Debate    | 94.92    | 62.12              |
> | GPT-4、GPT-3.5、GPT-3.5  | EoT-Report    | 94.84    | 62.43              |
> | GPT-4、GPT-3.5、GPT-3.5  | EoT-Relay     | 94.16    | 62.18              |
> | GPT-4、GPT-3.5、GPT-3.5  | EoT-Memory    | 94.31    | 63.42              |
> |                          |               |          |                    |
> | GPT-4、GPT-4、GPT-3.5    | Majority Vote | 95.52    | 90.78              |
> | GPT-4、GPT-4、GPT-3.5    | EoT-Debate    | 96.66    | 120.69             |
> | GPT-4、GPT-4、GPT-3.5    | EoT-Report    | 96.81    | 120.05             |
> | GPT-4、GPT-4、GPT-3.5    | EoT-Relay     | 96.96    | 123.71             |
> | GPT-4、GPT-4、GPT-3.5    | EoT-Memory    | 96.59    | 124.18             |
> |                          |               |          |                    |
> | GPT-4、GPT-3.5、Claude-2 | Majority Vote | 95.75    | 66.56              |
> | GPT-4、GPT-3.5、Claude-2 | EoT-Debate    | 96.89    | 86.60              |
> | GPT-4、GPT-3.5、Claude-2 | EoT-Report    | 96.58    | 88.64              |
> | GPT-4、GPT-3.5、Claude-2 | EoT-Relay     | 97.04    | 86.33              |
> | GPT-4、GPT-3.5、Claude-2 | EoT-Memory    | 96.81    | 90.06              |
>
> > R2: Could you add more analysis about the relationship between the four modes and the explanation of their performance differences?
>
> Your question provided an insightful perspective. We continued our experiments with GPT-4, GPT-3.5, and GPT-3.5 but swapped the model positions. We observed that when a stronger model, like GPT-4, acts as the primary node in the Debate and Report modes, the performance significantly improves. In contrast, the decentralized Relay and Memory modes showed little change based on the ordering of the models. In the future, we intend to explore this further across more datasets and also factor in the number of nodes to understand the relationship between model count and performance in different modes. These experimental findings and subsequent analyses will be added to our paper. We are truly grateful for this valuable angle of inquiry, and we will delve deeper into these aspects.
>
> | Method     | Order                     | Accuracy |
> | ---------- | ------------------------- | -------- |
> | EoT-Debate | GPT-3.5、GPT-3.5、GPT-3.5 | 89.54    |
> | EoT-Debate | GPT-4、GPT-3.5、GPT-3.5   | 94.92    |
> | EoT-Debate | GPT-3.5、GPT-4、GPT-3.5   | 93.93    |
> | EoT-Debate | GPT-3.5、GPT-3.5、GPT-4   | 94.01    |
> |            |                           |          |
> | EoT-Report | GPT-3.5、GPT-3.5、GPT-3.5 | 89.69    |
> | EoT-Report | GPT-4、GPT-3.5、GPT-3.5   | 94.84    |
> | EoT-Report | GPT-3.5、GPT-4、GPT-3.5   | 93.78    |
> | EoT-Report | GPT-3.5、GPT-3.5、GPT-4   | 93.78    |
> |            |                           |          |
> | EoT-Relay  | GPT-3.5、GPT-3.5、GPT-3.5 | 89.70    |
> | EoT-Relay  | GPT-4、GPT-3.5、GPT-3.5   | 94.16    |
> | EoT-Relay  | GPT-3.5、GPT-4、GPT-3.5   | 94.08    |
> | EoT-Relay  | GPT-3.5、GPT-3.5、GPT-4   | 94.08    |
> |            |                           |          |
> | EoT-Memory | GPT-3.5、GPT-3.5、GPT-3.5 | 90.06    |
> | EoT-Memory | GPT-4、GPT-3.5、GPT-3.5   | 94.31    |
> | EoT-Memory | GPT-3.5、GPT-4、GPT-3.5   | 94.31    |
> | EoT-Memory | GPT-3.5、GPT-3.5、GPT-4   | 94.01    |
>
> References:
>
> [1] Bisht N, Singh S. Analytical study of different network topologies[J]. International Research Journal of Engineering and Technology (IRJET), 2015, 2(01): 88-90.

---

### Official Review · Reviewer_UcTV · 2023-08-04

**Soundness:** 4

**Excitement:**

4: Strong: This paper deepens the understanding of some phenomenon or lowers the barriers to an existing research direction.

**Paper Topic And Main Contributions:**

This paper proposes a novel multi-model prompting approach, which the authors term Exchange-of-Thought. The proposed method is based on the intuition that allowing multiple LLMs to exchange information about the reasoning procedure leading to a prediction should help the models correct their reasoning mistakes and, therefore, improve their performance. The authors propose four variants of EoT, each of which involves different information-sharing dynamics among the models. The EoT approach is compared to three other recent CoT-based prompting strategies (CoT with self-consistency, ComplexCoT, and Progressive hint Prompting), showing consistent improvements on nine math and commonsense reasoning datasets. The authors additionally present multiple analyses focusing on different aspects and components of their proposed method (e.g., the impact of the terminating condition and of the confidence evaluation, the number of communication rounds between the models, …).

**Questions For The Authors:**

In lines 203-204, you state that “Wang et al. (2023) introduce Exchange-of-Thought prompting that utilizes evidence triples to […]”. This is probably a writing mistake (?). The method introduced by Wang et al., 2023 is named Chain-of-Knowledge Prompting.


**Reasons To Accept:**

The paper proposes a multi-model prompting strategy based on the interesting (though not novel) idea of allowing multiple LLMs to exchange reasoning information to obtain a common answer to a query. The authors’ implementation of this idea is shown to be effective with a solid experimental procedure: EoT shows consistent gains over three popular prompting strategies on nine different datasets. The analyses provided in Section 5.3 are useful to understand the effectiveness of EoT.

**Reasons To Reject:**

One concern with the paper’s contribution regards the computational cost of the proposed method. It is not clear what is the additional computational cost required by EoT, in comparison to standard CoT or to the three CoT-based baselines. A fair comparison between single- and multi-model prompting strategies would involve taking into account the overall computational cost needed to obtain the predictions (e.g., in terms of length of context and number of tokens generated).

**Reproducibility:**

4: Could mostly reproduce the results, but there may be some variation because of sample variance or minor variations in their interpretation of the protocol or method.

**Reviewer Confidence:**

4: Quite sure. I tried to check the important points carefully. It's unlikely, though conceivable, that I missed something that should affect my ratings.

---

> ### Author Rebuttal · Authors · 2023-08-28
>
> Thank you very much for your positive feedback on our work and for recognizing the effectiveness of our approach. We appreciate your constructive suggestion regarding the computational cost analysis of EoT.
>
> To address your concern, we've calculated both input and output token counts as well as the computational costs on the GSM8K dataset. We determined the computational cost based on OpenAI's official pricing for GPT-3.5 Turbo 4K as: input tokens\*0.0015/1000+output tokens\*0.002/1000
>
> The table below showcases the accuracy, average input token count, average output token count, and total computational cost of the gpt-3.5-turbo-0301 model across various methods. Taking the EoT-Relay method as an example, its total cost is 5.92, which is comparable to CoT-SC(3) and Complex-CoT(1). However, the accuracy of EoT-Relay (89.70%) significantly outperforms both CoT-SC(3) at 83.85% and Complex-CoT(1) at 80.83%. When compared to methods with similar accuracies, such as CoT-SC(20) at 89.46% and Complex CoT-SC(9) at 89.15%, EoT-Relay's cost is substantially lower than CoT-SC(20)'s 35.16 and Complex CoT-SC(9)'s 47.09. Consequently, EoT can markedly enhance accuracy at a comparable computational cost and can also reduce computational costs at similar accuracy levels. Our results from other EoT methods also support this conclusion. We find this insight immensely valuable. Due to time constraints, we've only conducted experiments on the GSM8K dataset. Moving forward, we plan to run experiments on more datasets and will augment our paper with these results and analyses. Thank you again for your invaluable suggestion!
>
> We sincerely apologize for the typo in lines #203-204 and any confusion it may have caused. We appreciate your meticulous attention to detail and have already corrected this error. Once again, thank you for your invaluable feedback and assistance^_^
>
> |                   | Accuracy(%) | Avg Input Tokens (tokens/sample) | Avg Output Tokens (tokens/sample) | Computational Cost($) |
> | ----------------- | ----------- | -------------------------------- | --------------------------------- | --------------------- |
> | CoT(1)            | 79.53       | 743.70                           | 108.19                            | 1.76                  |
> | CoT-SC(3)         | 83.85       | 2231.10                          | 328.69                            | 5.28                  |
> | CoT-SC(5)         | 85.90       | 3718.50                          | 540.78                            | 8.78                  |
> | CoT-SC(9)         | 88.32       | 6693.30                          | 984.82                            | 15.84                 |
> | CoT-SC(20)        | 89.46       | 14874.01                         | 2173.83                           | 35.16                 |
> | Complex-CoT(1)    | 80.83       | 2478.70                          | 124.26                            | 5.23                  |
> | Complex CoT-SC(3) | 85.37       | 7436.10                          | 371.11                            | 15.69                 |
> | Complex CoT-SC(5) | 87.03       | 12393.50                         | 624.53                            | 26.16                 |
> | Complex CoT-SC(9) | 89.15       | 22308.30                         | 1120.36                           | 47.09                 |
> | PHP               | 85.10       | 6494.74                          | 382.57                            | 13.85                 |
> | EoT-Debate        | 89.54       | 2504.83                          | 384.17                            | 5.97                  |
> | EoT-Report        | 89.69       | 2528.01                          | 386.70                            | 6.01                  |
> | EoT-Relay         | 89.70       | 2479.60                          | 385.88                            | 5.92                  |
> | EoT-Memory        | 90.06       | 2530.79                          | 385.32                            | 6.02                  |

---

### Meta-Review · Area_Chair_28Cu · 2023-09-19

**Recommendation:** 4

**Metareview:**

The main strength of this paper lies in coming up with a simple yet effective idea (of multiple LLMs communicating their thoughts to each other) and demonstrating consistent gains across 9 datasets and multiple CoT baselines. The reviewers found the experiments extensive and the analysis valuable. Moreover, the additional data provided by the authors during the rebuttal period was helpful in clarifying some open questions/concerns; I would strongly suggest that the authors include them in the revised version of the paper as they have indicated they would like to do.

One concern about the work is that the idea of multiple agents or LLMs communicating is not novel. E.g., Decomposed Prompting (ICLR-2023), ReAct (ICLR-2023), etc., have considered solving compositional problems by composing steps from simpler individual agents. That said, the way communication across multiple agents is used in the proposed setup is sufficiently different from prior uses.

---

### Decision · Program_Chairs · 2023-10-07

**Decision:**

Accept-Main

**Comment:**

The main strength of this paper lies in coming up with a simple yet effective idea (of multiple LLMs communicating their thoughts to each other) and demonstrating consistent gains across 9 datasets and multiple CoT baselines. The reviewers found the experiments extensive and the analysis valuable. Moreover, the additional data provided by the authors during the rebuttal period was helpful in clarifying some open questions/concerns; I would strongly suggest that the authors include them in the revised version of the paper as they have indicated they would like to do.

One concern about the work is that the idea of multiple agents or LLMs communicating is not novel. E.g., Decomposed Prompting (ICLR-2023), ReAct (ICLR-2023), etc., have considered solving compositional problems by composing steps from simpler individual agents. That said, the way communication across multiple agents is used in the proposed setup is sufficiently different from prior uses.